# Primary care and cancer: an analysis of the impact and inequalities of the COVID-19 pandemic on patient pathways

Toby Watt  ,[1,2] Richard Sullivan,[3,4] Ajay Aggarwal[4,5]

[1]REAL Centre, The Health Foundation, London, UK
[2]Public Health, Environments and Society, Faculty of Public Health and Policy, London School of Hygiene and Tropical Medicine, London, UK
[3]School of Cancer and Pharmaceutical Sciences, King's College London, London, UK
[4]Department of Oncology, Guy's and St Thomas' NHS Foundation Trust, London, UK
[5]Health Services Research and Policy, Faculty of Public Health and Policy, London School of Hygiene and Tropical Medicine, London, UK

**Correspondence to**
Toby Watt;
toby.watt@health.org.uk

## ABSTRACT

**Objectives** We explore the routes to cancer diagnosis to further undertanding of the inequality in the reduction in detection of new cancers since the start of the pandemic. We use different data sets to assess stages in the cancer pathway: primary care data for primary care consultations, routine and urgent referrals and published analysis of cancer registry data for appointments and first treatments.

**Setting** Primary and cancer care.

**Participants** In this study we combine multiple data sets to perform a population-based cohort study on different areas of the cancer pathway. For primary care analysis, we use a random sample of 5 00 000 patients from the Clinical Practice Research Datalink. Postreferral we perform a secondary data analysis on the Cancer Wait Times data and the National Cancer Registry Analysis Service COVID-19 data equity pack.

**Outcome measures** Primary care: consultation, urgent cancer referral and routine referral rates, then appointments following an urgent cancer referral, and first treatments for new cancer, for all and by quintile of patient's local area index of multiple deprivation.

**Results** Primary care contacts and urgent cancer referrals in England fell by 11.6% (95% CI 11.4% to 11.7%) and 20.2% (95% CI 18.1% to 22.3%) respectively between the start of the first non-pharmaceutical intervention in March 2020 and the end of January 2021, while routine referrals had not recovered to prepandemic levels. Reductions in first treatments for newly diagnosed cancers are down 16.3% (95% CI 15.9% to 16.6%). The reduction in the number of 2-week wait referrals and first treatments for all cancer has been largest for those living in poorer areas, despite having a smaller reduction in primary care contact.

**Conclusions** Our results further evidence the strain on primary care and the presence of the inverse care law, and the dire need to address the inequalities so sharply brought into focus by the pandemic. We need to address the disconnect between the importance we place on the role of primary care and the resources we devote to it.

## Strengths and limitations of this study

► This study draws from multiple data sets along the complex, multidisciplinary cancer pathway.
► We use a rich primary care data set containing patient level primary care activity linked to patients' local area socioeconomic indicator.
► Our primary care patient sample is relatively small (500 000 active patients from January 2016 to January 2021); however, the data produces results that closely mirror the rates of consultation and urgent cancer referral per patient produced in publicly available national data sets.

## Key messages

► Primary care is key part of the pathway for early cancer diagnosis through both routine and 2-week wait referrals.
► Cancer diagnosis rates have experienced a sustained fall since the start of the COVID-19 pandemic and introduction of non-pharmaceutical interventions 'lockdowns'.
► The fall in urgent cancer referral is larger than the fall in primary care contacts, implying that the content of consultations has shifted away from potential cancer diagnosis.
► Despite having a smaller reduction in primary care contact through the pandemic, patients living in poorer areas have had larger reductions in urgent cancer referrals and first treatments for new cancer.
► Government, patients and primary care staff must work together to catch up on missing diagnosis.
► Resilience in primary care is key for the cancer diagnosis pathway and must be developed for future disruptions, particularly in poorer areas where care is more complex.

## INTRODUCTION

The COVID-19 pandemic has had a profound impact on UK's health system. Each part of the UK's National Health Service (NHS) has been impacted in different ways, and we are still feeling many of the consequences of both the COVID-19 pandemic and the public health measures put in place to manage it (non-pharmaceutical interventions, NPIs). Cancer is one of the most complicated diseases that the UK health system must manage, being responsible for over one in four UK deaths in

2019. Cancer outcomes are acutely sensitive to changes in social determinants, patient pathways and service provision. Delays in both diagnosis and treatment have significant impacts on patient outcomes.[1 2] Pandemic-related diagnostic delays, lack of capacity and downstream stage progression (to more advanced disease) are already being seen.[3] In addition, the impact of the pandemic needs to be seen in the context of an already overstretched UK cancer care system prepandemic that was 'burning hot' even in normal times.[4]

Primary care sits at the heart of the cancer patient pathway and is the most crucial interface for early diagnosis and referral to hospital-based care, in addition to their wider support of patient with undergoing and after treatment. As models of cancer care have evolved in light of both technical advances and an ageing comorbid population, primary care has become an increasingly important aspect of integrated cancer care and an expansion of general practitioner (GP) roles in cancer care.[5] On average, 22.5% of patients diagnosed with cancer are referred to oncology diagnostic services from primary care, but this reflects wide site-specific variation from as little as 8.3% of breast cancer to 42% for bladder cancer.[6]

It is important to reflect that prior to the start of the COVID-19 pandemic, primary care had seen significant declines in overall resourcing relative to the funding of the rest of the NHS and compared with growing levels of disease burden that is managed in primary care. In addition, there is growing evidence that primary care has been under greater pressure in more deprived areas, with higher levels of staff turnover,[7] higher levels of complex multimorbidity,[8] higher numbers of consultations[9] and lower levels of funding and fewer GPs per capita once levels of ill health are taken into account.[10] These pressures on primary care, and a desire to correct them, have been recognised in the NHS Long Term Plan.[11]

Thus, to understand the COVID-19's impact on primary care and the downstream impact on cancer outcomes we need to see that the pandemic arrived when the system that was already struggling to cope. Prior to COVID-19, the central role of primary care as agents of change in reducing inequalities had been the subject of much debate yet could do little in the face of political avoidance of health equity.[12] Primary care had become a mirror on inequalities but also subject to significant pressures from these growing inequalities that had put practices in deprived populations under significant stress. Yet despite this, equity-oriented primary care reform in England in the mid-to-late 2000s may have helped to reduce socioeconomic inequality in health.[13] (box 1)

It is now clear that the UK experience of the pandemic was one of the worst in the world, both in terms of excess mortality (both COVID-19 and non-COVID-19) and the impact of NPI (lockdowns) on both the ability of health services to continue provide care and the impact of messaging (stay at home) on patients' timely presentation for care.[14] However, the overwhelming focus of impact studies on cancer care has been on hospital-based

> **Box 1  Non-pahamaceutical interventions implemented in England in response to the COVID-19 pandemic**
>
> COVID-19 was officially declared a pandemic by the WHO on 11 March 2020, and the Government announced its first full lockdown in England and the wider United Kingdom on 23rd March. In the following months England's NPI were eased, schools reopened in phases, non-essential shops reopened and in August the population were encouraged to eat out. Some restrictions were re-imposed in September and October, on the 5th of November 2020 a second brief national locked lasted until 2nd December. On the 6th of January 2021, a third national lockdown was introduced.[53]

services, including diagnostics. Given primary care's central role in pathways to diagnosis and integrated cancer care, including survivorship, there has been little insight around how overall changes in consultation rates impacted both routine and 2-week wait referrals as well as how this varied both in terms of site-specific cancers and as a consequence of socioeconomic inequalities. This study aimed to analyse the socioeconomic inequalities in the impact of NPI measures taken in response to COVID-19 on consultations and routine and urgent cancer referrals in primary care and cancer diagnosis in secondary care.

## METHODS
### Study design, data sources and participants
We perform a population-based cohort study using the following three separate sources.

### Primary care data: CPRD Aurum
Primary care electronic health records were obtained from the Clinical Practice Research Datalink Aurum database (henceforth CPRD). We included patient records from 1 January 2016 to 31 January 2021. Prepandemic data were included to establish long-term trends and patterns of seasonality in primary care use and referrals to secondary care. Similar to recent analysis of the COVID-19 pandemic,[15] our analysis focuses on comparing observed levels of activity to the expected following the introduction of NPI in England in March 2020.

CPRD contains anonymised patient primary care data from approximately 7% of the UK population and is broadly representative in terms of age, sex and ethnicity.[16] The patient records include information on consultations, patient demographic information, diagnoses, medication prescriptions and referrals to secondary care.

The period of eligibility for study inclusion starts on the latest of the study start date (1 January 2016) or the patient's registration to their practice. A patient's period of eligibility ends on the earliest of leaving their practice, the end of data collection from their practice or their death. Primary care records from CPRD were linked to the deciled index of multiple deprivation (IMD) from 2015 (https://www.gov.uk/government/statistics/english-indices-of-deprivation-2015)[17] of each patient's lower layer super output area (geographic areas in

England and Wales that are built from groups of contiguous output areas and have been automatically generated to be as consistent in population size as possible, and typically contain from four to six output areas. The minimum population is 1000, and the mean is 1500. For more details visit: (https://www.datadictionary.nhs.uk/nhs_business_definitions/lower_layer_super_output_area.html#:~:text=Lower Layer Super Output Areas,statistics in England and Wales). About 500 000 patients were randomly sampled from the CPRD population in England who were eligible for linkage within the defined study period.

### Cancer wait time data
Cancer waiting time (CWT) measure performance against the NHS Constitution Standards, recording the number of patients screened, referred to oncology specialists, diagnosed and treated for cancer. These measures are used by local and national organisations to monitor the timely delivery of services to patients, and they are published quarterly by NHS Digital (https://www.england.nhs.uk/statistics/statistical-work-areas/cancer-waiting-times/).

### Cancer diagnosis by socioeconomic status: NCRAS cancer equity data
Data on cancer diagnosis by socioeconomic group were drawn from the Cancer Alliance Data, Evaluation and Anlysis Service (CADEAS) and National Cancer Registry Analysis Service (NCRAS) that have two published data sets,[18] presenting the latest national data on:
1. The number of urgent suspected 2-week wait referrals (http://www.ncin.org.uk/view?rid=4346 (accessed on 24 January 2022)).
2. First definitive treatments for cancer (http://www.ncin.org.uk/view?rid=4347 (accessed on 24 January 2022)).

These data packs are produced based on the CWT data, with analysis from Hospital Episode Statistics and other sources outlined in their technical notes (further details in online supplemental annex 1).

### Study outcomes
#### Primary care consultations
We define consultations in CPRD data by a set of rules developed based on two variables in the consultations file (https://cprd.com/primary-care) ('EMIS consultation source identifier' and 'Consultation source code identifier') (These variables contain strings that categorise the patient record input and are selected by the staff member completing the record). In line with the approach taken by Carey *et al* 2012 for CPRD Gold data, we use a combination of the consultation code and the category of the record to identify consultations (details in online supplemental annex 2).

Using the observation file in CPRD Aurum, we were also able to identify where patients had influenza vaccinations. We look to exclude influenza vaccines from our analysis on the basis that the programme was expanded

in 2020/2021 to achieve maximum uptake (https://www.england.nhs.uk/wp-content/uploads/2020/05/Letter_AnnualFlu_2020-21_20200805.pdf). To help with the comparability of consultations in the two periods, we removed primary care appointments that included a influenza vaccine.

### Referrals from primary care: routine and urgent cancer
Referrals in CPRD are categorised into routine and 'urgent cancer'. Referrals from the 'referral file' are linked to patients, and no additional data cleaning steps were taken in the analysis of referrals.

### First appointment following an urgent referral
The CWT data present monthly counts of patients in England who have been recorded as receiving a first appointment following an urgent referral from primary care. The CWT data record this because the NHS has a 2-week performance target (online supplemental annex 3).

The NCRAS cancer equity data contain monthly counts in England of appointments following an urgent cancer referral broken down by tumour type and by deprivation according to patient's place of residence.

### First treatment following a cancer diagnosis
The CWT data present monthly counts of patients in England who have been recorded as receiving a first treatment for a new cancer diagnosis. The CWT data record this because the NHS has a 31-day performance target (online supplemental annex 3).

The NCRAS cancer equity data contain monthly counts in England of first treatments for new cancer broken down by tumour type and by deprivation according to patient's place of residence.

### Patient and public involvement
No patients involved.

### Data analysis
#### CPRD and CWT
For both CPRD and CWT, we separate the data into two, before and after the introduction of the first NPI.

Our analysis of CPRD primary care is conducted weekly and split into two periods before and after the introduction of NPI on 23 March 2020 (pre-NPI data are from 3 January 2016 to 21 March 2020, and our post-NPI onset data are from 22 March 2020 to 30 January 2021).

CWT data are reported monthly, our pre-NPI data are therefore from 1 October 2009 to 31 March 2020 and our post-NPI onset period is from 1 April 2020 to 31 January 2021.

We perform a linear regression of consultations, urgent and routine referrals from CPRD data and appointments following an urgent cancer referral and first treatments from CWT data over time to estimate expected values for the post-NPI onset period, based on predicted values from the data pre-NPI. To account for seasonality and time trends, we include

months as a categorical variable and time as a continuous variable, the approach taken by Carr *et al*.[15] In the case of weekly primary care data, we observe large dips in activity in weeks that include bank holidays and include a categorical variable on the basis of the number of bank holidays in each week (in the winter holidays in England there is always 1 week with two bank holidays). Our primary care activity rates are presented per 100 000 patient-months (We adjust the weekly rates per active patient in our sample to 100 000 patient-months: weekly rate per registered patient in sample× 100 000× (52/12)). When analysing primary care consultation rates by socioeconomics, we adjust for population age. We do so when calculating the consultation rates by IMD quintile and weighting the sample according to the European Standard Population (https://www.causesofdeath.org/docs/standard.pdf).

### NCRAS equity data

The analysis presented in the equity data pack compares new instances of first treatments in months during the pandemic (1 April 2020–31 January 2021) compared with the same months in 2019/2020. The analysis includes a 95% CI for the changes, based on rate ratios under an assumption that the population is the same in the pre-COVID-19 baseline and COVID-19 months. This is calculated using the exact method described in Breslow & Day 1987, pp 93-95.[19] The NCRAS equity data pack shows the high levels of heterogeneity in the impact of the COVID-19 pandemic on different tumour locations. The NCRAS data equity pack is different in its count and analysis of 'all tumours' compared with the Cancer Wait Times Data, and this is because the data are slightly different (cleaned and analysed by NCRAS—details in online supplemental annex 1). Results of our analysis with each data set are compared in online supplemental annex 4. Our presentation of these data follows the same method but presents the cumulative difference for the period from April 2020 to the end of January compared with the previous 12 months.

## RESULTS
### Overall impact of the pandemic

In the calendar year of 2019, before the COVID-19 pandemic and the associated NPI, there was an average of 39 127 primary care consultations per 100 000 patient-months. This equates to 4.70 attended appointments per registered patient or an estimated 266 million appointments in primary care nationally in 2019 (For comparison, the NHS national appointments data recorded 272 million attended appointments in primary care in 2019. Found here: https://digital.nhs.uk/data-and-information/publications/statistical/appointments-in-general-practice/march-2021).

Primary care consultations (figure 1A) dropped rapidly to a low of 26 919 consultations per 100 000 patient-months in the week following 29 March 2020, and this was 66.0% lower than the predicted rate. Rates slowly recovered over the next 24 weeks and by 5 September 2020 were up to 99% of the baseline. In total, there were an estimated 19.7 million (95% CI 19.5 to 20.0) fewer primary care consultations in the English NHS during this period. Primary care consultations again fell to below 90% of predicted levels in the third-wave NPI starting on 6 January 2021, and by the end of January 2021 there were a further 6.4 million fewer consultations than expected. Between the start of the first NPI in March 2020 and the end of January 2021, there were an estimated 26.1 million (95% CI 25.7 to 26.5) fewer appointments than expected (table 1A).

In 2019, the average rate of urgent cancer (2-week wait) referral was 314 per 100 000 patient-months, equating to an estimated 2.12 million for the NHS in England. Following the first NPI, urgent cancer referrals from primary care (figure 1B) fell to a nadir of 86 per 100 000 patient-months by 29 March 2020 (29.7% of the predicted

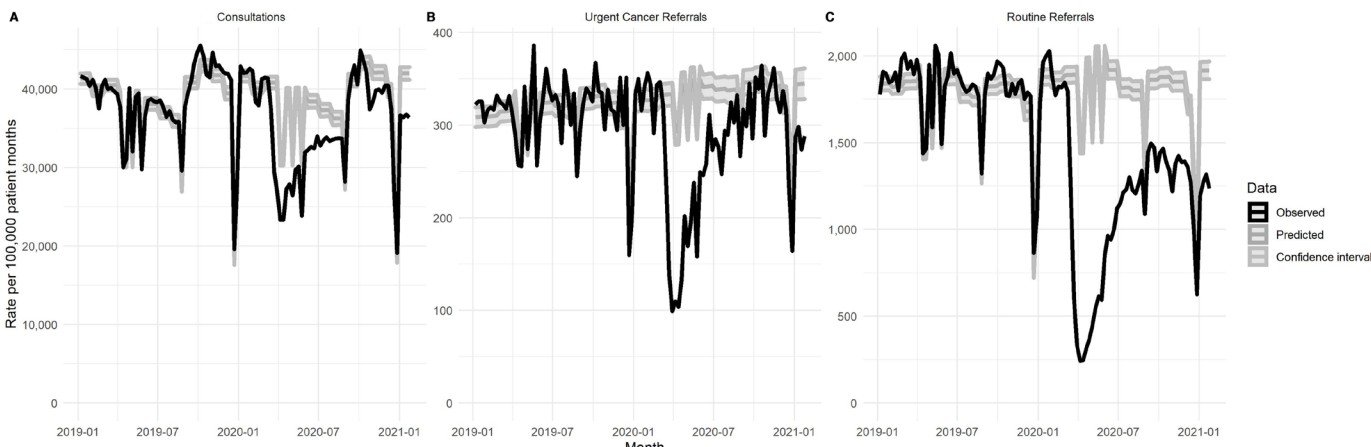

**Figure 1** Observed versus expected primary care activity, 1 January 2019–30 January 2021 (per 100 000 patient-months) (Clinical Practice Research Datalink Aurum data). (A) Consultations, (B) urgent cancer (2-week wait) referrals from primary care and (C) routine referrals from primary care.

**Table 1A** Observed post-COVID-19 primary care activity (CPRD Aurum), 22 March 2020–30 January 2021

| | Observed rate | Expected rate | Percentage reduction (95% CI) | Estimated no. missing from England population, to three significant digits (95% CI) |
|---|---|---|---|---|
| Event rate per 100 000 patient-months | | | | |
| Consultations | 34 201 | 38 684 | 11.6 (11.4 to 11.7) | 26 100 000 (25 700 000 to 26 500 000) |
| Routine referrals | 1067 | 1812 | 41.1 (40.4 to 41.8) | 4 330 000 (4 210 000 to 4 460 000) |
| Urgent cancer (2-week wait) referrals | 268 | 336 | 20.2 (18.1 to 22.3) | 395 000 (344 000 to 446 000) |

CPRD, Clinical Practice Research Datalink.

level). Urgent cancer referrals did not return to prepandemic baseline until the week following 23 August 2020 equating to 317 000 (95% CI 280 000 to 356 000) estimated lost urgent cancer referrals over this period. There was a second fall in urgent cancer referrals from primary care in the winter to below 90% of the baseline following the third lockdown (164 referrals per 100 000 patient-months in the week beginning 27 December 2021). This resulted in a further estimated 91 705 fewer urgent cancer referrals than expected. Between the start of the first NPI in March 2020 and the end of January 2021, there were 395 000 (95% CI 344 000 to 446 000) fewer urgent cancer referrals than expected (table 1A).

Routine referrals however have shown a different trajectory in that their rates did not recover to prepandemic levels (figure 1C). As a share of predicted levels, routine referrals had the greatest fall, dropping to 16.1% of predicted rates in the week from 19 April 2020. From then to the end of January, the closest it came to predicted levels was 80.3% in the week flowing 13 September 2020. For the 4 weeks in January 2021, it had fallen back down to 60%–70% of predicted rates. In 2019 there were an average of 1801 routine referrals per 100 000 patient-months from primary care, equivalent to an estimated 12.2 million for the NHS in England. Between the start of the first NPI in March 2020 and the end of January 2021, there were 4.33 million (95% CI 4.21 to 4.46) fewer routine referrals than expected (table 1A).

Patient demographics and patient-time and total numbers of observed consultations and routine and urgent referrals in our CPRD sample are presented in online supplemental annex 5.

Table 1A summarises the missing appointments and referrals for the postpandemic period. Since the start of the pandemic in March we have observed consultations rates that are 11.6% (95% CI 11.4 to 11.7) lower than predicted by previous data. The number of referrals to secondary care per consultation has also fallen, with urgent cancer referrals 20.2% (95% CI 18.1 to 22.3) and routine referrals 41.1% (95% CI 40.4 to 41.8) lower than expected.

The knock-on effect of the reductions in patients' primary care appointments and referrals can be observed in the national CWT data. The number of first

**Table 1B** Observed post-COVID-19 cancer diagnostic activity (Cancer Wait Times), 1 April 2020–31 January 2021

| | Observed rate | Expected rate | Percentage reduction (95% CI) | Estimated no. missing from England population, to three significant digits (95% CI) |
|---|---|---|---|---|
| Event rate per 100 000 patient-months | | | | |
| First consultant appointments following urgent referral from primary care | 296 | 366 | 19.2 (19.1 to 19.3) | 398 000 (395 000 to 401 000) |
| Incidence rate per 100 000 patient-months | | | | |
| First treatments for new cancer from the urgent primary care referral pathway | 21.4 | 25.5 | 16.1 (15.5 to 16.8) | 23 300 (22 200 to 24 400) |
| First treatments for new cancer from the national screening pathway | 1.63 | 3.47 | 53.2 (52 to 54.3) | 10 400 (10 000 to 10 900) |
| First treatments for new cancer | 39.7 | 47.4 | 16.3 (15.9 to 16.6) | 43 600 (42 500 to 44 700) |

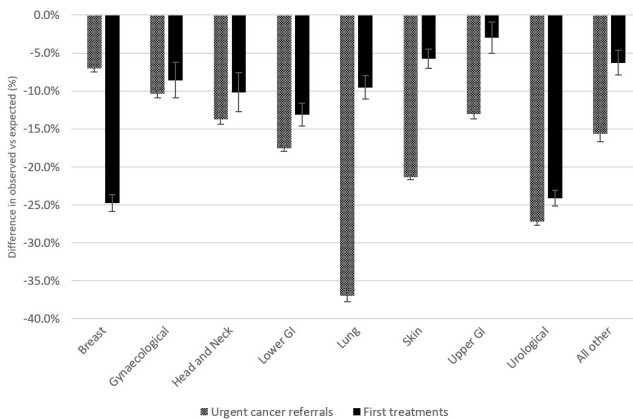

**Figure 2** Percentage difference between observed and expected first treatments for new cancer and urgent cancer referrals by tumour location from National Cancer Registry Analysis Service Cancer equity data pack (%, 1 April 2020 to 31 January 2021).

appointments with a cancer specialist following an urgent cancer referral has fallen by approximately the same amount as estimated for the referrals themselves: 19.2% (95% CI 19.1% to 19.3%). The number of cancer first treatments (following a diagnosis and decision to treat) was 16.3% (95% CI 15.9 to 16.6) lower than expected or 43600 (95% CI 42500 to 44700) missing first treatments from 1 April 2020 to 31 January 2021 (Dates for the CWT and NCRAS analysis do not line up with the CPRD analysis because the latter is conducted weekly, not monthly). (Graphs of observed compared with expected are presented in online supplemental annex 6).

Urgent cancer referrals by site-specific cancer from 1 April 2020 until 31 January 2021 showed significant heterogeneity from moderate reductions in urgent referrals for suspected breast (7.0%; 95% CI 6.6% to 7.5%) and gynaecological cancers (10.3%; 95% CI 9.7% to 10.9%) and greater reductions for lung (36.9%; 95% CI 36.1% to 37.8%) and urological (27.2%; 95% CI 26.7% to 27.7%) cancers (figure 2, further details in online supplemental annex 4, table A4.1). To show how pathway delays interface with reductions in cancer referrals we also examined reductions in first treatments for the same site-specific cancers over this period (figure 2). Breast and urological cancers observed the greatest reduction in new first treatments: breast fell by 24.8% (95% CI 23.6% to 25.9%) which equates to 10000 missing treatments and urological by 24.1% (95% CI 23.2 to 25.2) which equates to 12100 missing treatments. Taken together, these data reflect substantial delays in both diagnostic and treatment phases of the patient pathway.

### Inequalities in cancer diagnosis outcomes in the pandemic

There are inequalities in primary care use in England, with the people who live in the poorest areas have higher rates of consultation than those in richer areas once we adjust for age. The most deprived quintile was expected to have 43184 consultations per 100000 patient-months (table 2), 15% more than the least deprived.

The reduction of consultations over the period 22 March 2020 to 30 January 2021 was smallest for those in most deprived areas. Their reduction in consultations for the non-age-standardised figures was 9.6% (9.2%–9.9%), while for the least deprived the reduction was 12.4% (95% CI 13.2% to 13.9%) (table 2). Weekly levels of age-standardised consultations per 100000 patient-months by IMD quintile are presented in online supplemental annex 7.

Despite a smaller reduction in primary care contacts, we observe the largest reduction in both urgent cancer referrals and first treatments for cancer for patients living in the most deprived areas. The NCRAS data equity pack presents the number of urgent cancer referrals and first cancer treatments by IMD quintile (They do not age-standardise their results.). Figure 3 shows the reduction in urgent cancer referrals and first treatments for newly diagnosed cancer by IMD quintile.

There was a greater percentage reduction in urgent cancer referrals for those living in the most deprived areas in England, who experienced a 17.6% (95% CI 17.2% to 18.0%) reduction between 1 April 2020 and 31 January 2021 compared with the same period 12 months before, while referrals for the least deprived quintile fell by proportionately less: 15.3% (95% CI 14.9% to 15.6%). This equates to a reduction of 61500 referrals for the most deprived and 62600 or the least: without adjusting for age, the most deprived quintile had a smaller proportion of the prepandemic urgent cancer referrals, with 350000 referrals compared with 410000 for the least deprived quintile from April 2019 to January 2020.

At the same time, rates of new treatment for cancer for the people living in the most deprived 20% of England experienced a 15.8% (95% CI 14.6% to 17.0%) reduction between 1 April 2020 and 31 January 2021 compared with the same period 12 months before (6 610 missing first treatments). The reduction for the least deprived was 12.6% (95% CI 11.5% to 13.7%) which equates to 6880 missing first treatments.

Despite having more access to primary care for patients in more deprived areas (9.7% reduction for most deprived compared with 12.5% for the least deprived), urgent cancer referrals and newly diagnosed cancers have been disrupted by the pandemic more for people living in poorer areas.

### DISCUSSION

The coronavirus SARS-CoV-2 (COVID-19) pandemic has had a profound impact on the management of patients with cancer.[20] The first national lockdown in March 2020 created a ripple of NPIs, including 'stay at home' orders, diminished healthcare service provision and redistribution of healthcare to COVID-19-related care that has had a profound impact on cancer services.[1 21]

**Table 2** Observed post-COVID-19 primary care activity (CPRD Aurum) by IMD quintile, actual and age-standardised

| | 22 March 2020–30 January 2021 (weekly) | | |
| --- | --- | --- | --- |
| | Observed rate | Expected rate | Percentage reduction (95% CI) |
| Consultations per 100 000 patient-months | | | |
| IMD quintile—1 (least deprived) | 33 813 | 38 601 | 12.4 (12.1 to 12.7) |
| IMD quintile—2 | 34 169 | 38 793 | 11.9 (11.6 to 12.3) |
| IMD quintile—3 | 35 069 | 40 127 | 12.6 (12.3 to 12.9) |
| IMD quintile—4 | 33 494 | 37 793 | 11.4 (11 to 11.7) |
| IMD quintile—5 (most deprived) | 34 561 | 38 212 | 9.6 (9.2 to 9.9) |
| Consultations per 100 000 patient-months (age-standardised*) | | | |
| IMD quintile—1 (least deprived) | 32 927 | 37 636 | 12.5 (12.2 to 12.8) |
| IMD quintile—2 | 33 916 | 38 647 | 12.2 (11.9 to 12.6) |
| IMD quintile—3 | 35 535 | 40 870 | 13.1 (12.7 to 13.4) |
| IMD quintile—4 | 36 271 | 41 148 | 11.9 (11.5 to 12.2) |
| IMD quintile—5 (most deprived) | 38 997 | 43 184 | 9.7 (9.4 to 10) |

*Age standardisation is performed according to the European Standard Population.
CPRD, Clinical Practice Research Datalink; IMD, index of multiple deprivation.

There are also new potential barriers to the pathway that have resulted and may exacerbate these findings. For example, decreases in health-seeking behaviour due to the fear of acquiring COVID-19 infection through interactions with healthcare settings, increasing the use of remote consultations,[22] changes in routine referral guidelines,[23] as well as changes in the capacity of acute care. The backlog for routine diagnostic services is a particular concern given that approximately 40% of cancer are diagnosed through this route.[24]

Similar issues have also been identified within the health systems of other high-income countries. Primary care providers in eight European countries experienced similar issues in how to rapidly transform services in the wake to COVID-19.[25] A study in Sweden found an almost identical percentage reduction in primary care consultations (12%) as a result of the pandemic,[26] in Norway there was a 24% reduction in cancer referrals,[27] the

Netherlands had a 26% reduction in non-skin cancer diagnoses[28] and in Belgium there was a 44% reduction in diagnosis of invasive tumours in the first wave of the pandemic.[29] Our results do not appear to be unique to England: while different countries can have different routes to diagnosis,[30] many countries also observed disruptions to cancer pathways.[31–34]

While it was already known that there had been a substantial reduction in the number of overall cancer-related referrals,[35 36] the quantification of this had been missing. Our findings, that primary care consultations in English NHS fell by 12.4% between January 2020 and January 2021 with urgent cancer referrals even more suppressed (20.2%), reflect how profound the pathway disruptions were for patients with cancer. Furthermore, many cancers are picked up through the course of routine referrals from general practice for non-specific symptoms. The drop in routine referrals that we found (4.3 million, over this period) will inevitably translate into late-stage presentation and a substantial reduction in outcomes. This will include wider economic costs due to more expensive, late-stage treatment and productivity losses due to morbidity and premature mortality. However, the trajectory of the declines reflect not just changes to national policy in terms of NPI but also knock-on effects around public behaviour, primary care staffing, downstream reductions in diagnostics and an overall increase in friction across all cancer pathways and systems.

This reduction in cancer pathways through primary care needs to be put in the context of wider disruptions. The suspension of national cancer screening programmes meant that around 2 million people were not screened for cancer through national programmes.[37 38] Moreover, delays in cancer diagnoses and treatments have consistently been associated with poorer outcomes.[1 2]

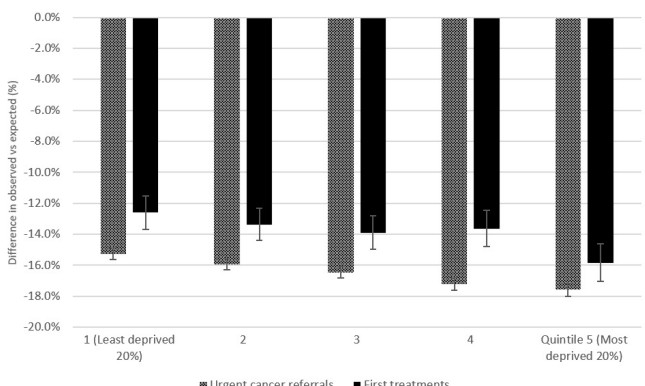

**Figure 3** Percentage difference between observed and expected urgent cancer referrals and first treatments for cancer by index of multiple deprivation quintile (1 April 2020–31 January 2021).

The COVID-19 pandemic has also exacerbated the worst 62-day CWT targets in the last decade where one of four patients urgently referred from primary care between April 2020 and January 2021 did not receive treatment within 62 days.[36]

In our analysis of urgent cancer referrals by site in relation to reductions seen in first treatments, significant differences were seen, which is also reflected in the international evidence. Urological cancers (testis, renal, prostate and urothelial) have been particularly impacted with greater than 25% decrease both in urgent referrals and first treatments. This suggests that outcomes will be particularly impacted in this group. Lung, skin and lower gastrointestinal (colon and rectal) cancer also experienced significant declines in urgent referrals; in the Netherlands, there was a 60% reduction in skin cancer diagnosis during the first wave.[28]

Breast cancer was the least impacted of all in terms of urgent referrals but experienced a 25% reduction in first treatments. This highlights how much breast cancer diagnosis relies on screening programmes which have suffered badly as a result of the pandemic in England[36] and internationally.[39] In England, head and neck cancers (HNC) saw a 10.2% (95% CI 7.6% to 12.7%) reduction in diagnosis, while studies in other geographies showed wide differences in the measures' impact of the pandemic on HNC: a study in Ontario, Canada, found no evidence of a reduction in HNC diagnosis following an initial drop in the 6 weeks following lockdown,[40] a clinic in Italy had just a 3.7% reduction in HNC,[41] a 14% reduction in Belgium,[29] a clinic in California showed a 22% reduction[42] and a Cancer Centre in the North of England reported a 33% reduction in HNC cases.[43] There is further international evidence of the impact of COVID-19 on interventions down the pathway, with reductions in radical cancer surgeries in two major cancer hubs in England and Italy.[44]

Differences in systems, populations and NPI from the pandemic present high levels of complexity in tackling the recovery at both national and local levels. Although it is possible that, in many countries, some patients with cancer have already been 'lost' to the system, that is, died of COVID-19 or other non-COVID-19 comorbidities, a significant number will now present with later stage disease, creating further pressure on acute cancer care.

Our findings also reflect socioeconomic inequalities, with more profound decrease in urgent cancer referrals and first treatments for the most deprived populations despite relatively better preservation of consultation rates. This is unexpected and extremely worrying, indicating greater disruption to the diagnostic pathway for patients living in more deprived areas, whose cancer outcomes were typically worse than their less deprived counterparts prepandemic.[45 46] Resilience in primary care is the key for cancer diagnosis pathway and must be developed. We know that there are challenges associated with resourcing health services in poorer areas (the inverse care law[47]), resulting in fewer resources per head of sick patient[10] and shorter consultation times.[48] Further research should focus on understanding to what extent complex morbidity, which is greater in poorer areas,[8 49] contributes to the disruption of the cancer diagnostic pathway. Greater understanding would help health systems better prepare for the kind of disruption we have seen as a result of COVID-19.

## Limitations

This study uses multiple data sets to analyse a complex and disjointed pathway. We include a primary care data set that uses a relatively small (500 000) patient sample. However, the CPRD data produce results that closely mirror the rates of consultation per patient (and their reduction) produced in NHS Digital's appointments data.[50] In addition, the estimated reduction in urgent cancer referrals is close to those presented in the NCRAS's analysis of their cancer registry data (tables 1A and 1B). It is not yet possible to link these data on a patient basis due to delays in data access and once possible further research would be illuminating.

## CONCLUSIONS

Our data reflect a disruption to a complex interaction of several systemic issues that place a great deal of impetus on the role of primary care in ensuring early diagnosis of cancer. Primary care was already under strain prepandemic, with low levels of investment and workforce deficits.[51] Particularly in areas of high deprivation, general practice is underfunded and under staffed relative to need.[7 8 10]

Early cancer diagnosis requires concordance of each participant and mechanism—including patients' awareness and ability to present with cancer symptoms, the ability of GPs to detect and urgently refer possible cancer cases and sufficient diagnostic capacity (in terms of both workforce and equipment) to enable swift referrals and minimise delays to diagnosis and treatment. Every one of these nodes on the pathway to early diagnosis has been affected by the pandemic and the national policy response. However, further work is required as there is currently little understanding and even less evidence about how much each disruption is ultimately impacting cancer pathways.

The impact of the pandemic on cancer diagnosis and time to treatment shown here is very serious. However, what is more concerning is the unequal and inequitable impact on those worst off. Cancer as a disease area 'magnifies what we know to be true about the totality of the health care system. It exposes all its strengths and weaknesses'.[52] Our results further evidence the strain on primary care, the presence of the inverse care law[47] and the dire need to address the inequalities so sharply brought into focus by the pandemic. We need to address the disconnect between the importance we place on the role of primary care in cancer care and the resources we devote to it.

**Contributors** TW, RS and AA designed the study. Data acquisition, cleaning and analysis was conducted by TW on the Health Foundation's secure date environment. TW wrote the manuscript in the first instance. TW, RS and AA interpreted the data and substantially reviewed the draft manuscript. All authors approved the final version of the manuscript. TW and RS are the guarantors.

**Funding** This publication is funded through the UK Research and Innovation GCRF grant ES/P010962/1.

**Competing interests** None declared.

**Patient and public involvement** Patients and/or the public were not involved in the design, conduct, reporting or dissemination plans of this research.

**Patient consent for publication** Not required.

**Ethics approval** CPRD collect data for research use. We did not require ethical approval; however, scientific approval for this study was given by the CPRD Independent Scientific Advisory Committee (20_143).

**Provenance and peer review** Not commissioned; externally peer reviewed.

**Data availability statement** Data are available in a public, open access repository. Data may be obtained from a third party and are not publicly available. The primary care activity data may be obtained from a third party and are not publicly available. We used deidentified primary care data from the Clinical Practice Research Datalink (CPRD). For more information, please visit: https://www.cprd.com/Data-access, and enquiries can be emailed to enquiries@cprd.gov.uk. Scientific approval for this study was given by the CPRD Independent Scientific Advisory Committee (ISAC). The study was approved by the ISAC for CPRD research (20_143). The data are provided by patients and collected by the NHS as part of their care and support. Other data sources are available in a public, open access repository: Cancer Wait Times at https://www.england.nhs.uk/statistics/statistical-work-areas/cancer-waiting-times/ and the NCRAS Cancer data equity pack is available at http://www.ncin.org.uk/local_cancer_intelligence/cadeas.

**ORCID iD**
Toby Watt http://orcid.org/0000-0002-0217-0228

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
