## [Reviewer comments · BMJ Open]

ARTICLE DETAILS

TITLE (PROVISIONAL)	Primary Care and Cancer: an analysis of the impact and inequalities of the COVID-19 pandemic on patient pathways
AUTHORS	Watt, Toby; Sullivan, Richard; Aggarwal, Ajay

VERSION 1 – REVIEW

REVIEWER	Laing, Scott University of Ottawa, Department of Family Medicine
REVIEW RETURNED	11-Dec-2021

GENERAL COMMENTS	Thank you for the opportunity to review your work. This is quite an impressive amount of data and an extensive analysis that you have performed! It really provides some valuable insight into the current state of cancer diagnosis and wait times in relation to the volume of primary care service delivery. There are some great findings in your work that are important for policy makers and health system leaders to know when addressing the backlog that has developed. I did have some questions about clarity and some suggestions for improvements, which are outlined below. I encourage a revision and resubmission of this important work. The comments are extensive, so I apologize as I know lengthy reviews are daunting, but these are intended to improve the quality of your work. I attached a PDF with highlights to help you in revisions and I think most comments translate between the two. Thank you, 1. Is the research question or study objective clearly defined? Page 5, line 1-2: While you have provided a good description of the work that you have done, I did not get a clear sense of the study aim. It would help to have an aim statement explicitly written, ie "This study aimed to..." 4. Are the methods described sufficiently to allow the study to be repeated? Yes, based on the descriptions and the figures I think I could reproduce this work. However, I did have some questions about the analysis. - Why do the start times vary in the analysis? Table 1a is March 15, 2020. Table 1b is April 1, 2020. Table 2 is March 22, 2020. Can you make the dates consistent? - What was adapted from the Carey et al 2012 CPRD Gold data methodology? I couldn't see this in the Annex or body, could you provide a brief description of what was different?
--

	- Why were influenza vaccination consultations excluded? How do you know that only influenza vaccination was provided based on these search terms? Many primary care providers will address multiple issues in a consultation, so this may have excluded many consultations where other services were provided. Noted that the time periods of vaccinations shown align with normal vaccination time periods. I also am unclear on what data was being searched for the exclusion terms and what the cutoff was for exclusion (one word present, five present, all present). A brief clarification of both would help. - I note in Table 1a it states that the Estimated # missing from England population, but also state CPRD is a UK database. Was this a typo? Or did you restrict your sample to only England residents? 6. Are the outcomes clearly defined? Reading through the methods I wasn't entirely clear what the outcomes were. I think there could be added clarity about what the outcomes are. Example page 5 line 45 change "CPRD Aurum" subheading to something more meaningful, like "Primary Care Consultations". Same goes for other outcome sections. 7. If statistics are used are they appropriate and described fully? The estimations provided appear to not follow significant digit rules. This provides an overly specific estimate, which is likely inaccurate. As well, since this is an estimate, it would help to see a confidence interval on the final estimates. Example, in Table 1a, 26.9 million (X to Y million; 95% CI). Noted as well, there were inconsistencies in the significant digits in the body of the manuscript as well. See attached PDF as I have highlighted ones that I saw. 9. Do the results address the research question or objective? Yes, but I think there are some consistency and clarity issues which were outlined above. If these are addressed, then I think the article will be much stronger and more clearly link the study aim, outcomes, and findings. 10. Are they presented clearly? The unit of analysis changes, which makes it challenging to see the parallel between the data sets. The unit of analysis magnitude changes with some using the per person level and other using the per 100 000 persons level. This changing scale makes it harder to understand the data being presented. Adjusting these to a consistent level would make the data more comparable. There are also inconsistencies where some figures use "per person-week", "per patient-week", "per person per week". I recommend to choose one phrasing and to explain a briefly how this was calculated or what it means. The table titles also state "Observed post COVID-19...", I am not sure what "post COVID" means. The pandemic is still very active globally and there wasn't mention that these are people that all had been diagnosed with COVID-19. Recommended to rephrase for clarity.  - Figures 2 and 3 are missing Y-axis labels. For clarity it would help to add context to percentage (ie, percent change) - Note Annex 5 isn't referenced in the results or discussion. Should it be? - In Annex 2, page 22 line 22 there is a missing table reference "Table Y", which table number does this refer to?
--	--

	- Page 7 line 16: it states "(03/03/2020 – 29/02/2020). This equates to 254 million appointments in primary care nationally." The dates appear to be incorrect here. For clarity it would help to state the timeframe that the 254 million consultations occurred in. 11. Are the discussion and conclusions justified by the results? Generally yes, there are a few points in the conclusion paragraph (page 12 line 14-17) and key messages (page 15 points 1, 5, 6) which are not in the discussion sections. These are relevant and important topics to discuss. If possible, it would be great to see them flushed out more in the discussion if these are the points the author would like to make. As well, the discussion about social 14. To the best of your knowledge is the paper free from concerns over publication ethics (e.g. plagiarism, redundant publication, undeclared conflicts of interest)? I am uncertain about Annex 1. It appears to be data that was already analysed elsewhere as it is discussed in the introduction, but nowhere else. I looked at the reference, but only found excel data tables. Was this part of the analysis? If so it should be included in the methods and results. If not part of the analysis the authors completed and is from an external source, then this is likely sufficient to be cited or the authors should confirm that they have permission to reproduce this table in their work.
--	--

REVIEWER	Trecca, Eleonora M IRCCS Ospedale Casa Sollievo della Sofferenza
REVIEW RETURNED	22-Dec-2021

GENERAL COMMENTS	Thank you for the opportunity to review this interesting manuscript entitled "Primary Care and Cancer: an analysis of the impact and inequalities of the COVID-19 pandemic on patient pathways". The article is fluent and well written, being the Authors native English language speakers. Moreover, it highlights the difficulties in primary care and especially in the treatment of oncological patients evidenced during the COVID-19 pandemic in UK but that unfortunately were similar to those experienced in many other countries. Since the management of these patients is time sensitive, it is very important to discuss this topic and release international strategies and guidelines that can be useful in the post-COVID-19 era. Methods and sources are appropriate; results are clearly presented. My suggestion is to improve the discussion, which appear too short, with further analysis:  - Are there in the current literature any other articles describing the impact of the COVID-19 pandemic on the management of ORL COVID-19 patients? - Could you make a comparison with other countries? How did they face the pandemic and the consequent reorganization of the system and counselling of cancer patients? For example, Italy has a national Healthcare system and it was one of the first countries to experience a large-scale outbreak. Please, find for references some articles discussing this topic: 10.1186/s13027-021-00352-9; 10.14639/0392-100X-N0941; 10.1007/s00405-020-06046-z; https://doi.org/10.3390/cancers13071597.
--

VERSION 1 – AUTHOR RESPONSE

Reviewer: 1

Dr. Scott Laing, University of Ottawa

Comments to the Author:

*** Please find additional comments from this reviewer in the attached file ***

We are incredibly grateful for the detail of your review and the care and attention paid to our manuscript. The marked-up pdf is particularly helpful, allowing us to accurately adapt the manuscript to incorporate your comments.

Thank you for the opportunity to review your work. This is quite an impressive amount of data and an extensive analysis that you have performed! It really provides some valuable insight into the current state of cancer diagnosis and wait times in relation to the volume of primary care service delivery. There are some great findings in your work that are important for policy makers and health system leaders to know when addressing the backlog that has developed.

I did have some questions about clarity and some suggestions for improvements, which are outlined below. I encourage a revision and resubmission of this important work. The comments are extensive, so I apologize as I know lengthy reviews are daunting, but these are intended to improve the quality of your work. I attached a PDF with highlights to help you in revisions and I think most comments translate between the two.

Thank you,

1. Is the research question or study objective clearly defined?

Page 5, line 1-2: While you have provided a good description of the work that you have done, I did not get a clear sense of the study aim. It would help to have an aim statement explicitly written, ie "This study aimed to..."

This is a great point; on page five we have amended the statement to be clearly assigning the aim of the research.

4. Are the methods described sufficiently to allow the study to be repeated?

Yes, based on the descriptions and the figures I think I could reproduce this work. However, I did have some questions about the analysis.

- Why do the start times vary in the analysis? Table 1a is March 15, 2020. Table 1b is April 1, 2020. Table 2 is March 22, 2020. Can you make the dates consistent?

Thank you for pointing this out. We should have been clearer both in the methods and the presentation. We are making use of patient level data and publicly available aggregate data. The former is most accurately performed weekly to line up with the introduction of lockdowns in the UK. Unfortunately, the publicly available data reporting cancer outcomes is recorded monthly. Table 1a was a typo, with results actually reflecting the period from March 22nd. Table 1b is the monthly data. Corrections have been made on this basis.

- What was adapted from the Carey et al 2012 CPRD Gold data methodology? I couldn't see this in the Annex or body, could you provide a brief description of what was different?

Thank you for this we have added clarification to the outcomes section on your recommendation. We have included descriptions of the two consultation variables in a foot note. The Carey et al approach is recommended by CPRD for analysis of consultations in CPRD Gold (a different database). There is

no agreed recommended method for Aurum, we have mirrored the Carey method however in that we have used both the consultation description variables and the category of the staff member who filled out the patient record in tandem to identify consultations. Manuscript has been amended to reflect this.

- Why were influenza vaccination consultations excluded? How do you know that only influenza vaccination was provided based on these search terms? Many primary care providers will address multiple issues in a consultation, so this may have excluded many consultations where other services were provided. Noted that the time periods of vaccinations shown align with normal vaccination time periods. I also am unclear on what data was being searched for the exclusion terms and what the cutoff was for exclusion (one word present, five present, all present). A brief clarification of both would help.

This is a very good point. We understand that the administration of flu vaccines in UK primary care can be conducted either by a GP during a consultation or by a member of primary staff separately. Our aim was to exclude "non-consultations", the reviewer has helpfully pointed out that we may be excluding too much. To our knowledge no one else has tried to distinguish between consultation with flu vaccines and separate flu vaccines using primary care data. There is therefore no scientific reference for how many consultations we would be excluding. We have therefore elected to amend the research to include the flu vaccines for completeness. The results relating to consultations change throughout the manuscript as a result, however our interpretation and broad results remain the same.

- I note in Table 1a it states that the Estimated # missing from England population, but also state CPRD is a UK database. Was this a typo? Or did you restrict your sample to only England residents? Thank you for spotting this. The study is based on the English population, we have added that detail to the data section which was accidentally omitted.

6. Are the outcomes clearly defined?

Reading though the methods I wasn't entirely clear what the outcomes were. I think there could be added clarity about what the outcomes are. Example page 5 line 45 change "CPRD Aurum" subheading to something more meaningful, like "Primary Care Consultations". Same goes for other outcome sections.

This is a helpful comment, we have rewritten the whole section to clarify the outcomes included and their data sources.

7. If statistics are used are they appropriate and described fully?

The estimations provided appear to not follow significant digit rules. This provides an overly specific estimate, which is likely inaccurate. As well, since this is an estimate, it would help to see a confidence interval on the final estimates. Example, in Table 1a, 26.9 million (X to Y million; 95% CI). Noted as well, there were inconsistencies in the significant digits in the body of the manuscript as well. See attached PDF as I have highlighted ones that I saw. We agree with the reviewer that we are being spuriously precise in our reporting. We have adjusted the results of the population extrapolated results to be to three significant figures and we have added confidence intervals to our national estimates.

9. Do the results address the research question or objective?

Yes, but I think there are some consistency and clarity issues which were outlined above. If these are addressed, then I think the article will be much stronger and more clearly link the study aim, outcomes, and findings.

We agree, your comments have made this into a much stronger manuscript.

10. Are they presented clearly?

The unit of analysis changes, which makes it challenging to see the parallel between the data sets. The unit of analysis magnitude changes with some using the per person level and other using the per 100 000 persons level. This changing scale makes it harder to understand the data being presented. Adjusting these to a consistent level would make the data more comparable. There are also inconsistencies where some figures use "per person-week", "per patient-week", "per person per week". I recommend to choose one phrasing and to explain a briefly how this was calculated or what it means.

Thank you for making this point, we agree it lacks clarity. Given that we are presenting results on cancer diagnosis (small numbers) we have elected to adjust all results to be presented as per 100,000 patient-months. We have added a description of how this was calculated to the methods section.

The table titles also state "Observed post COVID-19...", I am not sure what "post COVID" means. The pandemic is still very active globally and there wasn't mention that these are people that all had been diagnosed with COVID-19. Recommended to rephrase for clarity.

We agree, we have changed the phrasing to post-non pharmaceutical intervention (NPI) onset.

- Figures 2 and 3 are missing Y-axis labels. For clarity it would help to add context to percentage (ie, percent change)

Thank you, we have amended figures 2 and 3.

- Note Annex 5 isn't referenced in the results or discussion. Should it be?

Yes, thank you for pointing this out. Annex 5 includes more detail on the primary care population demographics. We have amended the results section accordingly.

- In Annex 2, page 22 line 22 there is a missing table reference "Table Y", which table number does this refer to?

Thank you for catching this, we now signpost to the correct table (now Table A1.2)

- Page 7 line 16: it states "(03/03/2020 – 29/02/2020). This equates to 254 million appointments in primary care nationally." The dates appear to be incorrect here. For clarity it would help to state the timeframe that the 254 million consultations occurred in.

Thank you, we agree this is unclear. This passage is designed to give a sense of the scale of primary care provision pre-pandemic and to compare our sample estimates to the national data. We have now adjusted this to cover the full year in 2019.

11. Are the discussion and conclusions justified by the results?

Generally yes, there are a few points in the conclusion paragraph (page 12 line 14-17) and key messages (page 15 points 1, 5, 6) which are not in the discussion sections. These are relevant and important topics to discuss. If possible, it would be great to see them flushed out more in the discussion if these are the points the author would like to make. As well, the discussion about social
Thanks you for pointing this out. We have moved our points made (page 12 line 14-17) to the discussion section. We are very happy to draw out the key messages more in the manuscript. This helpful comment will make for a much better paper.

14. To the best of your knowledge is the paper free from concerns over publication ethics (e.g. plagiarism, redundant publication, undeclared conflicts of interest)?

I am uncertain about Annex 1. It appears to be data that was already analysed elsewhere as it is discussed in the introduction, but nowhere else. I looked at the reference, but only found excel data tables. Was this part of the analysis? If so it should be included in the methods and results. If not part of the analysis the authors completed and is from an external source, then this is likely sufficient to be cited or the authors should confirm that they have permission to reproduce this table in their work.

Thank you for this comment, we agree and have removed the table presented in Annex 1 as per your recommendation.

Reviewer: 2

Dr. Eleonora M Trecca, IRCCS Ospedale Casa Sollievo della Sofferenza

Comments to the Author:

Thank you for the opportunity to review this interesting manuscript entitled “Primary Care and Cancer: an analysis of the impact and inequalities of the COVID-19 pandemic on patient pathways”.

The article is fluent and well written, being the Authors native English language speakers. Moreover, it highlights the difficulties in primary care and especially in the treatment of oncological patients evidenced during the COVID-19 pandemic in UK but that unfortunately were similar to those experienced in many other countries. Since the management of these patients is time sensitive, it is very important to discuss this topic and release international strategies and guidelines that can be useful in the post-COVID-19 era.

Methods and sources are appropriate; results are clearly presented.

My suggestion is to improve the discussion, which appear too short, with further analysis:

- Are there in the current literature any other articles describing the impact of the COVID-19 pandemic on the management of ORL COVID-19 patients?

Thank you for this consideration. We would have liked to have included more detail on different tumour locations. Unfortunately, we are restricted in our presentation of types of cancer by the analysis made available by the NCRAS – we are not able to get more detailed information than they provide.

- Could you make a comparison with other countries? How did they face the pandemic and the consequent reorganization of the system and counselling of cancer patients? For example, Italy has a national Healthcare system and it was one of the first countries to experience a large-scale outbreak. Please, find for references some articles discussing this topic: 10.1186/s13027-021-00352-9; 10.14639/0392-100X-N0941; 10.1007/s00405-020-06046-z; <https://doi.org/10.3390/cancers13071597>.

We are very grateful to reviewer number 2 for their comments. In response to reviewer 1’s comments we have expanded the discussion section to build on the key messages. Reviewer 2’s suggested papers were very helpful and prompted us to discover some similar work that reports disruption to cancer and primary medical services as a result of the pandemic. We have brought this matter into our discussion.

Reviewer: 1

Competing interests of Reviewer: I have read and understood BMJ policy on declaration of interests and declare that I have no competing interests.

Reviewer: 2

Competing interests of Reviewer: No conflict of interests.

VERSION 2 – REVIEW

REVIEWER	Laing, Scott University of Ottawa, Department of Family Medicine
REVIEW RETURNED	12-Feb-2022
GENERAL COMMENTS	Thank you for the opportunity to review your work again.

	I see significant improvements from the last manuscript providing improved clarity and statistical accuracy. In particular, thank you for making the unit of analysis consistent across the study and using significant digits, this has significantly improved the understandability of your work. As well, thank you for adding in confidence intervals, this provides a better understanding of certainty of your analysis. I recommend this manuscript for acceptance. Great work! ----- PS, for final proofs, there were two typos that I noticed: - page 4, line 17 where I believe the authors meant to write "evolved" instead of involved. - page 10, line 37 "covid-19" instead of "COVID-19"
--	--

REVIEWER	Trecca, Eleonora M IRCCS Ospedale Casa Sollievo della Sofferenza
REVIEW RETURNED	14-Feb-2022

GENERAL COMMENTS	Discussion could have been better improved according to my suggestions
--

VERSION 2 – AUTHOR RESPONSE

Reviewer: 1

Dr. Scott Laing, University of Ottawa

Comments to the Author:

Thank you for the opportunity to review your work again.

I see significant improvements from the last manuscript providing improved clarity and statistical accuracy. In particular, thank you for making the unit of analysis consistent across the study and using significant digits, this has significantly improved the understandability of your work. As well, thank you for adding in confidence intervals, this provides a better understanding of certainty of your analysis.

I recommend this manuscript for acceptance.

Thanks very much for your help, we agree it is a much better piece of work.

Great work!

PS, for final proofs, there were two typos that I noticed:

- page 4, line 17 where I believe the authors meant to write "evolved" instead of involved.
- page 10, line 37 "covid-19" instead of "COVID-19"

Reviewer: 2

Dr. Eleonora M Trecca, IRCCS Ospedale Casa Sollievo della Sofferenza

Comments to the Author:

Discussion could have been better improved according to my suggestions

Reviewer 2 original comments

Reviewer: 2

Dr. Eleonora M Trecca, IRCCS Ospedale Casa Sollievo della Sofferenza

Comments to the Author:

Thank you for the opportunity to review this interesting manuscript entitled “Primary Care and Cancer: an analysis of the impact and inequalities of the COVID-19 pandemic on patient pathways”.

The article is fluent and well written, being the Authors native English language speakers. Moreover, it highlights the difficulties in primary care and especially in the treatment of oncological patients evidenced during the COVID-19 pandemic in UK but that unfortunately were similar to those experienced in many other countries. Since the management of these patients is time sensitive, it is very important to discuss this topic and release international strategies and guidelines that can be useful in the post-COVID-19 era.

Methods and sources are appropriate; results are clearly presented.

My suggestion is to improve the discussion, which appear too short, with further analysis:

- Are there in the current literature any other articles describing the impact of the COVID-19 pandemic on the management of ORL COVID-19 patients? Added a short international comparison to the discussion from the ORL / ENT / Head and Neck cancer literature.

- Could you make a comparison with other countries? How did they face the pandemic and the consequent reorganization of the system and counselling of cancer patients? For example, Italy has a national Healthcare system and it was one of the first countries to experience a large-scale outbreak.

Please, find for references some articles discussing this topic: 10.1186/s13027-021-00352-9;

10.14639/0392-100X-N0941; 10.1007/s00405-020-06046-

z; <https://doi.org/10.3390/cancers13071597>.

We have further expanded the discussion section while endeavouring to keep the word count as low as possible. We have expanded the discussion of international reductions in cancer referrals and diagnoses as a whole. We found similarly worrying reductions in outcomes across many European countries.

We have taken on reviewer 2's comments and expanded the international discussion to include a comparison of the impact of COVID-19 on ORL (included in head and neck cancer in our study) in different geographies. We find that different national and local health systems showed widely different outcomes in different cancer types. This serves to highlight the complexities of the recovery and improving cancer diagnosis and care in different systems, for different cancers: there is unlikely to be a “one size fits all” solution.

It is a challenge to compare results on the impact of COVID-19 on cancer diagnosis because of these complexities. We believe that, as more and more research on the reduction and delay in cancer treatment is produced, a systematic review of national cancer care systems and the impact of NPIs from the pandemic on cancer would be extremely valuable.

VERSION 3 – REVIEW

REVIEWER	Trecca, Eleonora M IRCCS Ospedale Casa Sollievo della Sofferenza
REVIEW RETURNED	26-Feb-2022

GENERAL COMMENTS	accept as it is
-----------------